# Activation and friction in enzymatic loop opening and closing dynamics

Kirill Zinovjev[1], Paul Guénon [1,2], Carlos A. Ramos-Guzmán [1,3], J. Javier Ruiz-Pernía [1], Damien Laage [2] & Iñaki Tuñón [1,2] ✉

Protein loop dynamics have recently been recognized as central to enzymatic activity, specificity and stability. However, the factors controlling loop opening and closing kinetics have remained elusive. Here, we combine molecular dynamics simulations with string-method determination of complex reaction coordinates to elucidate the molecular mechanism and rate-limiting step for WPD-loop dynamics in the PTP1B enzyme. While protein conformational dynamics is often represented as diffusive motion hindered by solvent viscosity and internal friction, we demonstrate that loop opening and closing is activated. It is governed by torsional rearrangement around a single loop peptide group and by significant friction caused by backbone adjustments, which can dynamically trap the loop. Considering both torsional barrier and time-dependent friction, our calculated rate constants exhibit very good agreement with experimental measurements, reproducing the change in loop opening kinetics between proteins. Furthermore, we demonstrate the applicability of our results to other enzymatic loops, including the M20 DHFR loop, thereby offering prospects for loop engineering potentially leading to enhanced designs.

Enzymes are complex and flexible structures that can adopt different conformations necessary for their function[1,2]. Conformational changes occur during the enzymatic catalytic cycle and are often required to accommodate the substrate in the active site, position the catalytic residues correctly for the chemical transformation, and release the reaction product into the bulk. In some cases, these conformational transitions are the slowest step in the catalytic cycle, limiting the enzymatic turnover. Therefore, the mechanism and the factors governing the dynamics of enzymatic conformational changes have attracted considerable attention in recent years[3–7].

One of the most ubiquitous conformational changes observed during the catalytic cycles of enzymes is the closing and opening motions of loops that cover active sites. Loop motion is essential for catalysis in many natural enzymes, such as Dihydrofolate Reductase[8], Triosephosphate Isomerase[9] or Orotidine 5′-Monophosphate Decarboxylase[10], to name a few well-known examples. Loop closing

over active sites ensures substrate sequestration from the solvent[11] and improves active site preorganization, favoring its desolvation[12]. Experimentally, the structures of the open and closed conformations are accessible via X-ray crystallography[13], while the loop opening and closing kinetics can be measured by NMR relaxation[14,15]. Loop motions are found to occur on a broad range of timescales, typically from picoseconds to milliseconds[16], and in some cases are the rate-limiting step during the catalytic cycle. There is now growing evidence highlighting the tremendous importance of flexible loop dynamics not only for the regulation of enzymatic activity[17] but also for selectivity and thermal stability[17,18], and controlling the properties of flexible loops is a promising and attractive avenue to obtain enzymes with tailored features[17].

However, this requires determining the microscopic factors governing both the equilibrium between open and closed forms of enzymatic loops and the kinetics of these interstate conversions, and

[1]Departamento de Química Física, Universidad de Valencia, 46100 Burjasot, Spain. [2]PASTEUR, Département de Chimie, École Normale Supérieure, PSL University, Sorbonne Université, CNRS, 75005 Paris, France. [3]Instituto de Materiales Avanzados, Universidad Jaume I, 12071 Castelló, Spain. ✉e-mail: ignacio.tunon@uv.es

the latter have so far remained elusive. This is largely due to the complexity of large collective displacements occurring during the loop motions and the delicate balance between protein-protein, protein-solvent, and protein-ligand interactions that are involved. Some insights into the underlying factors controlling protein conformational dynamics can be gained from experimental and numerical studies of protein folding. Some studies have suggested that structural dynamics can be described as a diffusive motion on a rough energy landscape with friction caused both by the solvent viscosity and by intrachain protein interactions[19–22]. In contrast, temperature-jump experiments[23,24] suggest that loop conformational changes can also present large activation energies. Elucidating the molecular factors governing protein-loop dynamics thus requires identifying the molecular rearrangements responsible for the suggested barrier and the origin of friction.

The paradigm flexible loop protein that we have selected to investigate this issue is human Protein Tyrosine Phosphatase 1B (PTP1B). It is part of the Protein Tyrosine Phosphatases (PTPs) superfamily of enzymes whose activity is regulated by conformational loop motions[25]. PTP1B is involved in the regulation of insulin and leptin signaling and the signaling of epidermal growth factor[26]. It catalyzes the dephosphorylation of one of the tyrosine residues of its protein substrates in a two-step process involving the cleavage of the tyrosine phosphate monoester, followed by the hydrolysis of the phosphoenzyme intermediate. In the first step, the thiol group of a conserved cysteine (Cys215) acts as a nucleophile, breaking the phosphate bond to a tyrosine residue of the substrate and forming a thiophosphate enzyme intermediate. In the second step, this intermediate is hydrolyzed thanks to the nucleophilic attack performed by a water molecule[6]. Both steps are assisted by an aspartic residue (Asp181) that acts as a general acid/base and lies on a loop known as the WPD-loop, named for the three residues placed in the N-terminal side and conserved in the superfamily, Trp179-Pro180-Asp181 in PTP1B. The WPD-loop is a flexible Ω-loop consisting of a dozen residues (117–188), including the catalytic Asp181. Although this loop can exist in both open and closed conformations (see Fig. 1), only the closed-loop form allows catalysis[27]: the loop must be closed to bring the catalytic Asp181 into proximity with the substrate in the active site. Both open and closed forms have been observed in the apo and holo forms of the protein[25,28,29]. In fact, the substrate can bind to both forms[27]. As seen in Fig. 1, Asp181 forms a salt bridge with Arg112 in the open state, while in the apo closed state, the sidechain of Asp181 is rotated to establish a new interaction with Arg221, a residue of the active site that participates in substrate recognition.

The loop opening and closing rate constants have been determined experimentally using NMR techniques. In the apo form of PTP1B, the values obtained for $k_{closed}$ and $k_{open}$ are 22 and 890 s$^{-1}$ respectively, resulting in an equilibrium constant of 40 in favor of the open form[25]. A comparative study between PTP1B and the *Yersinia* PTP (YopH) found that the rates of loop motions mirror the catalytic rate constants in these two enzymes, the rate of loop motions in YopH being about 50 times larger than for PTP1B[25]. However, the barriers found for the chemical steps show only modest differences, suggesting that loop motions contribute to the observed differences in the catalytic rate constants between these two PTPs[6]. Kamerlin, Hengge, and coworkers built a series of chimeric enzymes with varied compositions of the WPD-loop, demonstrating that point mutations along the loop can alter the equilibrium between the open and closed states, changing the hydrogen bonding network established by the loop[27], and affecting the enzymatic activity[30]. In some cases, the increased mobility of chimeric enzymes can result in the exploration of unproductive conformations and consequently in a reduced catalytic rate constant[31]. These results thus stress the key role played by loop dynamics in PTP1B catalytic activity.

While these previous studies have already highlighted important factors contributing to the PTP WPD-loop conformational equilibrium[6,27,30,32], a detailed description of the mechanism governing the conformational change of enzymatic loops is still lacking. In this paper, our goal is thus to characterize this mechanism and its rate-limiting step, to identify the molecular origin of the large activation energy measured in temperature-jump experiments, and to establish the roles of solvent-induced and internal frictions, which have been shown to be important for protein folding dynamics[21]. We combine all-atom molecular dynamics (MD) simulations with a string-method approach to determine the complex reaction coordinate for the open/closed transition and critically, to identify the transition state (TS). Our simulations show that WPD-loop opening in PTP1B results from two successive processes: an activated and localized conformational change first occurs in the loop backbone, followed by diffusive loop motions. The key rate-limiting conformational change is the torsion of a single peptide group involving the Asp181 and Phe182 residues. Our analysis of the friction on the reaction coordinate relies on the Grote–Hynes theory[33] to describe the different response timescales in the system and reveals that the relevant friction exclusively arises from other torsions along the loop backbone which need to be rearranged when the key dihedral angles switch. This strong friction leads to a dynamical caging effect which considerably slows the barrier crossing. Finally, we show that our picture highlighting the importance of single

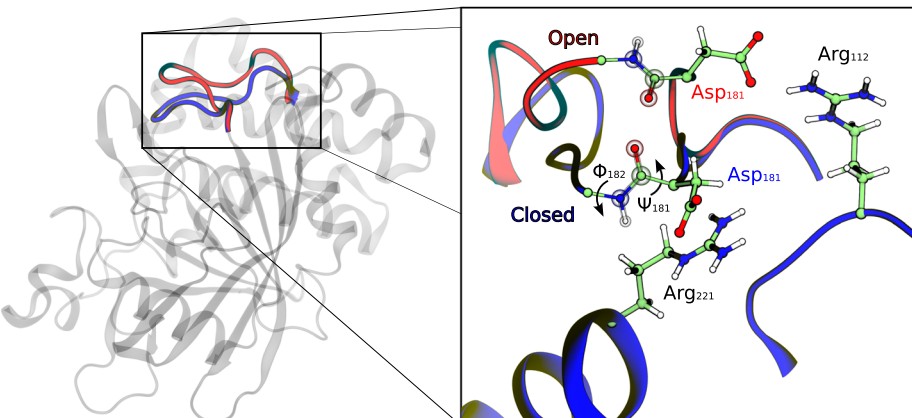

**Fig. 1 | Open and closed states of the WPD-loop in PTP1B.** Superposition of the open (red) and closed (blue) loop conformations of PTP1B from PDB structure 6B90. On the right a closer view of the loop highlighting the positions of Asp181 and the salt-bridges formed by this latter residue in the open (red) and closed (blue) states with Arg112 and Arg221 respectively. The dihedral angles $\Phi_{182}$ and $\Psi_{181}$ controlling the rotation of the peptide bond between residues 181 and 182 are also shown.

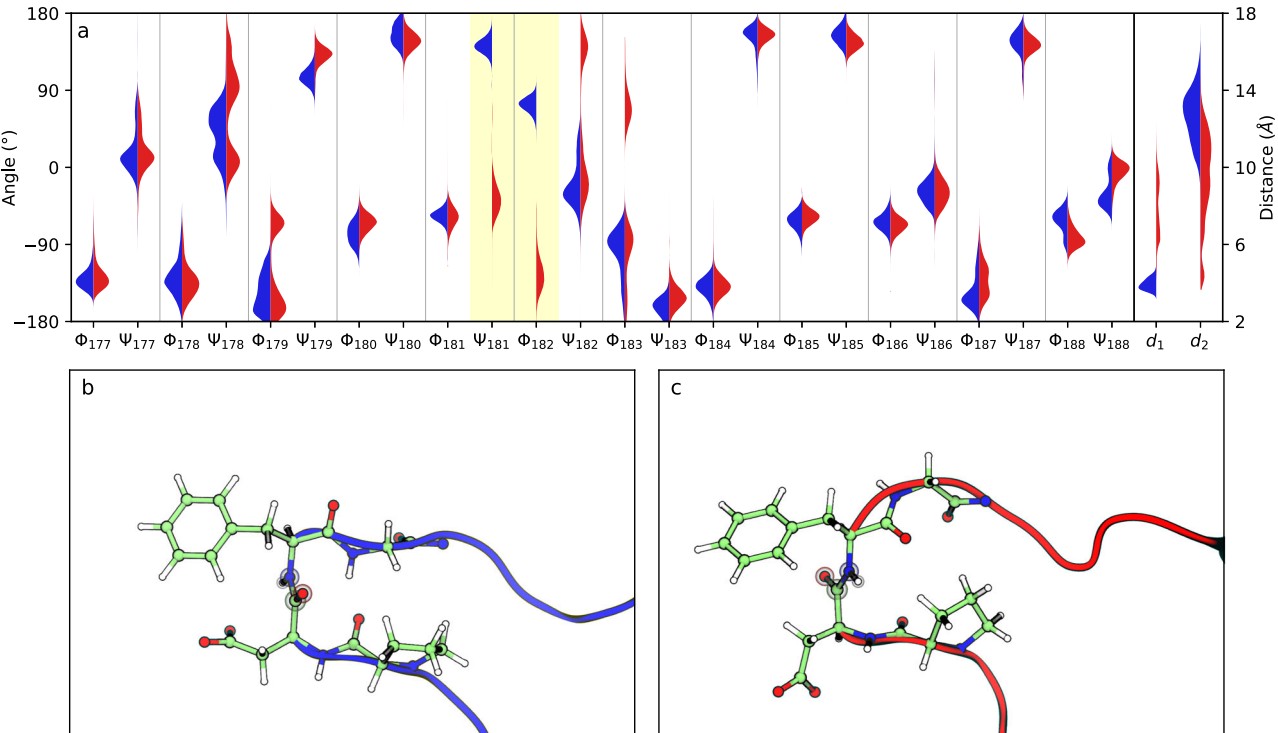

**Fig. 2 | Characterization of the open and closed states of the WPD-loop.**
**a** Probability distributions of the backbone dihedral angles (ϕ and ψ) corresponding to the WPD-loop obtained from 10 × 100 ns MD simulations of the open (red) and closed (blue) states along with the distances between the Cγ atom of Asp181 and the Cζ atoms of Arg221 ($d_1$) and Arg112 ($d_2$). The two dihedral angles that clearly distinguish the open and closed states are $\psi_{181}$, $\phi_{182}$ (highlighted in pale yellow). **b** Conformation of the Asp181-Phe182 peptide bond in the closed conformation. **c** Conformation of the Asp181-Phe182 peptide bond in the open conformation.

peptide group rotation strongly coupled to other torsions of the loop for opening and closing kinetics applies not only to other members of the PTP family but also extends to other types of enzymes including *E. coli* Dihydrofolate Reductase. This demonstrates the general applicability of our picture, offering new avenues for engineering flexible loop dynamics in proteins.

## Results

### MD simulations of the open/closed state and identification of order parameters

We ran 10 independent 100 ns long MD simulations for both open and closed states of the PTP1B apo form to analyze the order parameters that distinguish the two states. Running relatively shorter simulations from uncorrelated structures is, in general, a good strategy to sample correctly the configurational space of a given state[34]. We monitored all the ψ and ϕ torsion angles of the WPD-loop backbone as well as two distances that correspond to the salt-bridges formed in the open (Asp181Cγ-Arg112Cζ) and closed (Asp181Cγ-Arg221Cζ) states, see Fig. 1. The values of these distances in the X-ray open/closed conformations are 4.3/11.5 and 7.8/4.0 Å, respectively[28].

Figure 2 displays the probability distributions of the distances and torsion angles for simulations in the closed and open states. The analysis of the distances reveals that large transient displacements of the loop are possible, especially for the open state, and that the probability distributions of these distances for the two states overlap. Therefore, it is not possible to distinguish between these two conformational states using only these distances or a function of them (see Supplementary Fig. 1). In contrast, two torsion angles $\psi_{181}$ and $\phi_{182}$ clearly differentiate between both states because their distributions are clearly separated, as shown in Fig. 2 and Supplementary Fig. 1. The difference between the two conformational states in terms of these

torsions can be explained by the fact that the WPD-loop contains a β-turn defined by four residues, Pro180-Asp181-Phe182-Gly183. When the loop is closed, this β-turn adopts a standard type II conformation[35], stabilized by a hydrogen bond between the carbonyl group of Pro180 and the NH group of Gly183 (the N-O distance is 2.68 Å in the X-ray structure of the closed form, see Supplementary Fig. 2). In the open form, the peptide group between Asp181-Phe182 is rotated, as seen in Fig. 2b, c, with the amide and carbonyl groups pointing in opposite directions to those in the closed state and the Gly183N-Pro180O distance is now substantially larger, 5.22 Å in the X-ray structure. In a recent study, Shaw and coworkers also identified this motif as the key conformational switch in the closed-to-open transformation[32].

The previous analysis reveals that both closed and open states are stable, well separated and that the open-closed transition of the WPD-loop involves two distinct types of motions: (i) the displacement of the loop, which is reflected in the varying distances between Asp181 and the two anchoring arginines, and (ii) the conformational change of the backbone, particularly the peptide bond between residues Asp181-Phe182. Figure 2 and Supplementary Fig. 1 illustrate the different nature of these motions, with broad fluctuations observed in terms of distances, while the torsional angles $\psi_{181}$ and $\phi_{182}$ clearly differentiate the two states, indicating that changes in these dihedral angles are related to a free energy barrier between the open and closed forms. Before addressing the selection of an appropriate reaction coordinate to obtain the free energy profile corresponding to the loop conformational change, it is interesting to understand the reasons for the conformational change of the Asp181-Phe182 peptide bond. As mentioned earlier, the loop closes to attain a catalytically active conformation where the Asp181 residue forms a salt-bridge interaction with Arg221. The formation of this interaction necessitates the rotation of the Asp181-Phe182 peptide bond to prevent repulsion between the

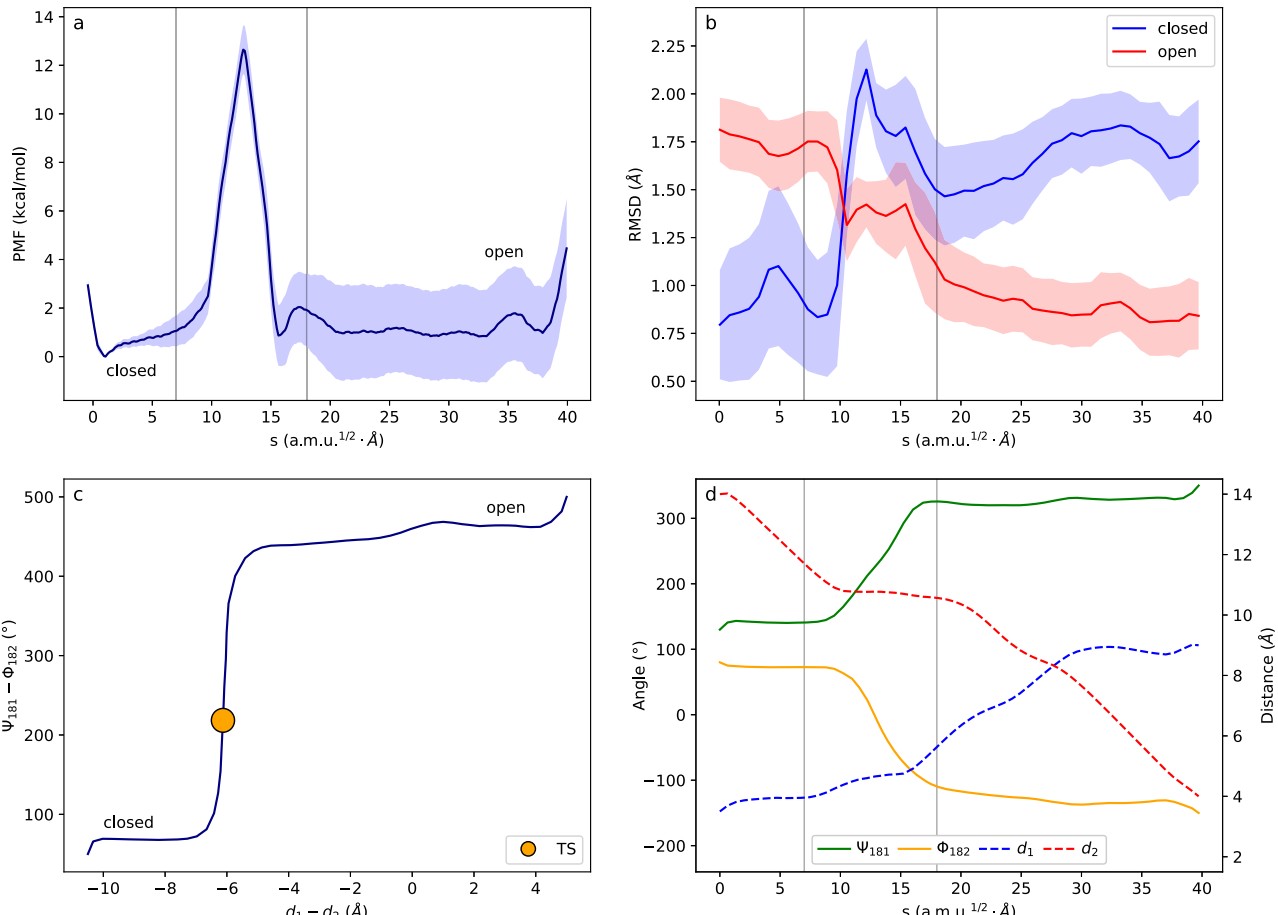

**Fig. 3 | Free energy landscape for the WPD-loop closed-to-open conformational change in PTP1B. a** Free energy profile for the closed (left) to open (right) transition along the *s* path-CV. The shaded region corresponds to the statistical uncertainty; **b** Average RMSD measured for the backbone atoms of the WPD-loop for snapshots obtained from the Umbrella Sampling simulations along the path-CV with respect to the X-ray structures (PDB 6B90) corresponding to the closed (blue) and open (red) states. The shaded region corresponds to the statistical uncertainty

(95% confidence interval); **c** Projection of the MFEP along the antisymmetric combinations of the two distances and two dihedral angles used as CVs. The yellow dot indicates the position of the Transition State; **d** Evolution of the individual CVs (distances on the right vertical axis and dihedrals on the left vertical axis) along the MFEP. The CVs used in the ASM calculations are: $\psi_{181}$, $\phi_{182}$ torsional angles and the distances Asp181C$\gamma$-Arg221C$\zeta$ ($d_1$) and Asp181C$\gamma$-Arg112C$\zeta$ ($d_2$). Vertical lines indicate the stages of the process discussed in the text.

carbonyl group and the carboxyl oxygen atoms of Asp181. This repulsion is eliminated when the peptide bond is rotated, and the carbonyl group is positioned towards the inner part of the loop, far from the carboxylate group (see Fig. 1). Thus, based on MD simulations, it has been observed that the open-closed transitions of the WPD-loop involve not only a displacement but also a significant conformational rearrangement.

**Reaction coordinate and free energy profile for the open/closed transition**

From our previous analysis, it is clear that the $\psi_{181}$ and $\phi_{182}$ torsional angles are key ingredients of a dividing surface separating the open and closed states and thus of a putative reaction coordinate. Although the salt-bridge distances between Asp181 and Arg112/Arg221 are not sufficient to distinguish between the two states, they can still be important to verify that the transition leads to the correct structures, particularly in the case of the open state where a wide range of structures exists[6]. Thus, we explored the free energy landscape for the open/closed transition using four collective variables (CVs): $\psi_{181}$, $\phi_{182}$, Asp181C$\gamma$-Arg221C$\zeta$ ($d_1$) and Asp181C$\gamma$-Arg112C$\zeta$ ($d_2$). We employed the Adaptive String Method (ASM)[36] as explained in the "Methods" section. This method determines the Minimum Free Energy Path (MFEP) for the transition along these four CVs and then builds a single path-CV

(denoted as *s*) to obtain a one-dimensional free energy profile using Umbrella Sampling (see "Methods" section).

Figure 3 shows the results of applying the ASM to study the open/closed transition of the WPD-loop in PTP1B. Figure 3a shows the Potential of Mean Force (PMF) along the path-CV (G(*s*)). To demonstrate that this method correctly drives the protein from the closed-loop to the open-loop conformation, Fig. 3b shows the average RMSD of the loop backbone atoms between structures sampled along the path and the reference X-ray structures of the closed and open states. A movie of the conformational change along the path is provided as Supplementary Movie 1. The G(*s*) profile displays two rugged free energy minima that correspond to the closed (left) and open (right) conformations with a sharp free energy barrier of ~12 kcal·mol⁻¹ between them. The barrier at $s^{\ddagger}$ separates the closed ($s < s^{\ddagger}$) and open ($s > s^{\ddagger}$) configurations. The equilibrium constant between the open and closed forms is determined by the free energy difference between the two states. We emphasize that the latter is not the free energy difference between the most stable geometries in each form but includes a contribution due to the number of configurations accessible in each form. While G(*s*) in Fig. 3a is determined by the probability of each specific value of the coordinate s, the relative free energies of the open and closed states are determined by the sums of the probabilities over the range of *s* values corresponding to the definition of each state.

With our reaction coordinate, the valley corresponding to the open form is significantly wider than that of the closed conformation. This can be seen as an entropic contribution to the free energy because the closed state is more constrained than the open state. The equilibrium constant between open and closed states thus depends on the integrated probabilities over the $s$ ranges defining each state, as described in the "Methods" section. The value obtained for the equilibrium constant from our free energy profile is 1.0, indicating that both states are equally probable. In contrast, the experimental value is 40[25], which translates into a free energy difference of 2.3 kcal·mol$^{-1}$, the open state being more stable than the closed state. The difference is within the uncertainty of our simulations (see Fig. 3a).

Figure 3c, d represent the projection of the path on the antisymmetric combination of distances and torsional angles and the evolution of the individual CVs along the MFEP, respectively. These figures show that the closed-to-open transition can be decomposed into three stages. This allows identifying the transition mechanism and the rate-limiting step. The first and the third stages essentially correspond to a change in the distances describing the salt bridges between Asp181-Arg112 and Asp181-Arg221. In the transition from the closed-to-open state, the Asp181-Arg221 salt bridge must first be broken, and finally, a new salt bridge, Asp181-Arg112, must be formed. These processes take place within each of the two free energy valleys, showing that these salt bridges can be formed or broken with small free energy changes, roughly within 2.0 kcal·mol$^{-1}$. The disruption of the salt bridges is facilitated by the presence of water molecules that can efficiently shield the charge-charge interaction between Asp181 and the arginine, modulating the energy gain associated to this interaction. This can be seen in the evolution of the number of solvent molecules around the carboxylate group of Asp181, that shows a clear increase in the intermediate stages of the process (see Supplementary Fig. 3).

The second stage of the closed-to-open transition is associated to the change in the $\psi_{181}$ and $\phi_{182}$ torsional angles. These angles change concertedly, with the first angle increasing and the second angle decreasing. This coordinated motion allows for the complete rotation of the Asp182-Phe182 peptide group (see Fig. 1). Once the Asp181-Arg221 interaction is broken, the peptide group can be rotated, resulting in the system transitioning to the open state. The open state is then stabilized by the formation of the new Asp181-Arg112 salt bridge. This second stage is responsible for the free energy barrier observed in Fig. 3a and, subsequently, for the rate of the process as discussed below.

According to the picture obtained from our MFEP calculations, loop opening in PTP1B is a combination of two kinds of motions: the loop displacement and the loop backbone conformational rearrangement. The former occurs along a rugged free energy landscape without large energy barriers, while the latter is clearly activated. To confirm this picture, we performed free molecular dynamic simulations initiated from configurations selected from the closed state and where we rotated the peptide group between residues 181 and 182, changing the $\psi_{181}$ and $\phi_{182}$ angles from the closed state values to values corresponding to the open state (see SI for details). Then the system was evolved without any external bias and after 1 μs of simulation, 18 of the 20 trajectories resulted in a stable open loop conformation with a significant displacement from the closed position (see Supplementary Fig. 4a). None of the trajectories reverted to a stable closed state, confirming the existence of a significant free energy barrier. Analysis of the loop displacement shows a linear increase of the mean squared Asp181Cα-Gly220Cα distance with time (see Supplementary Fig. 4b, c), corresponding to a diffusive displacement with a diffusion coefficient of 1.18·10$^{-2}$ Å$^2$·ns$^{-1}$. Considering that, according to the X-ray structures, the Asp181Cα-Gly220Cα distance must increase by 4.4 Å from the closed-to-open conformation, a purely diffusive loop motion should be completed in approximately 800 ns. However, the experimental rate constant shows that the PTP1B loop opening/closing process takes

place in the millisecond/second timescale[25], indicating the presence of a free energy barrier separating the closed and open states, in agreement with the proposed PMF. Our free MD simulations demonstrate that flipping a single peptide group, the one between residues 181–182, is the key factor triggering the WPD-loop conformational change in PTP1B, and that the free energy barrier in between the closed and open states is largely associated with the rotation of this particular peptide group. Once this rotation is completed, the loop can diffusively evolve from the closed-to-open state. Note that temperature-jump studies on different enzymes have demonstrated that loop conformational changes can present a significant activation barrier, in agreement with our picture[23,24].

## Evaluation of the rate constant and the impact of friction in loop motion

According to our previous description, the inverse rate constant for loop opening motion should be obtained by combining the inverse rates constants for the loop displacement ($1/k_{dis}$) and for the conformational change associated to the torsion of a single peptide bond ($1/k_{conf}$):

$$\frac{1}{k_{closed/open}} = \frac{1}{k_{dis}} + \frac{1}{k_{conf}} \tag{1}$$

As discussed above, the displacement of the loop is a diffusive motion that takes place in the ns-μs timescale, while the conformational change involves a large associated free energy barrier and takes place in the ms-s timescale. Therefore, we can focus on the latter contribution. Dihedral rotations in general and protein backbone conformational changes in particular, can be modeled as the passage over a one-dimensional free energy barrier subject to the friction exerted by the environment[19,37,38]. This friction results, in principle, from the coupling of solvent and protein degrees of freedom with the reaction coordinate, in this case essentially defined by a combination of the $\psi_{181}$, $\phi_{182}$ torsional angles (see TS crossing in Fig. 3c). We can thus express the rate constant as the product of two terms: one term associated with the free energy barrier that can be derived from Transition State Theory (TST) and a transmission coefficient due to the friction:

$$k_{close/open} = \kappa \cdot k_{close/open}^{TST} \tag{2}$$

The TST rate constant can be obtained from the free energy barrier along the path-collective variable $s$ as described in the "Methods" section[39]. In the context of protein conformational changes, the transmission coefficient ($\kappa$) has been usually modeled with Kramers' theory, that considers that the progress along the reaction coordinate is delayed by frictional effects due to the coupling with other degrees of freedom[40]. A key assumption of this theory is that environmental dynamics are infinitely faster than barrier-crossing dynamics, resulting in a full friction from all environmental coordinates. However, in the present situation, only part of the environment can respond on the timescale of barrier crossing, resulting in significant deviations from the behavior expected by this theory[33,41]. In this study, we used the Grote−Hynes (GH) approach based on the generalized Langevin equation to include frequency-dependent frictional effects on the reaction rate constant, accounting for the different response timescales of environmental degrees of freedom (see SI for details)[33].

The rate constants corresponding to the opening and closing processes are reported in Table 1. The information required for their calculation is provided in Supplementary Table 6. The calculated rate constants closely match the experimental values, with a maximum difference of only one order of magnitude. This can be translated into an error of only 2 kcal·mol$^{-1}$ in the corresponding activation free

energies. The statistical uncertainties obtained for the opening and closing activation free energies (see SI) are of 1.1 kcal·mol$^{-1}$.

The observed agreement between experimental and calculated rate constants supports the theoretical framework chosen to describe the loop opening process. In our approach, the rate is determined by a combination of a free energy barrier and a friction term, which enters through the transmission coefficient and slows down the process. Notably, the effect of the friction on the rate constant is not negligible. The value of the transmission coefficient according to GH is κ = 0.17 (see Supplementary Table 6), which reduces the rate of the process by almost one order of magnitude with respect to the TST estimation. Although the value of κ is small, Kramers' prediction is significantly smaller ($κ_{Kr}$ = 0.05), indicating that this theory overestimates the effect of friction by ignoring its time dependence, as previously recognized[33,41]. In general, the GH theory yields values that align more closely with transmission coefficients obtained from molecular dynamics (MD) simulations compared to Kramers' approach[42]. The value predicted here by GH theory agrees with the calculated ratio of reactive trajectories estimated from the free trajectories initiated at the TS depicted in Supplementary Fig. 5 (13 out of 60 trajectories are reactive, *i.e.*, evolve from closed-to-open states, some of them undergoing recrossings). This ratio serves as a qualitative estimation of the transmission coefficient derived from MD simulations and is useful to stress the better performance of GH theory with respect to Kramers' approach.

The molecular origin of friction during protein folding processes has been long discussed in the literature. In general, both solvent and protein degrees of freedom contribute to this friction, but the extent of their participation seems to be case-dependent[19,21,38,43]. Our framework provides a strategy for a systematic analysis. The time-dependent

**Table 1 | Experimental and calculated values for the rate constants (in s$^{-1}$) and equilibrium constant corresponding to the opening/closing process of the WPD-loop in PTP1B and YoPH**

|         |       | $k_{opening}$ | $k_{closing}$ | $K_{eq}$ |
|---------|-------|---------------|---------------|----------|
| PTP1B   | Exp.  | 890           | 22            | 40       |
|         | Calc. | 470           | 470           | 1.0      |
| YoPH    | Exp.  | 42000         | 1240          | 34       |
|         | Calc. | 79000         | 610           | 130      |

friction and its power spectra for the conformational change of the PTP1B WPD-loop are presented in Fig. 4. The friction exerted by the environment on the reaction coordinate is so high (the friction at $t = 0$ equals 960cm$^{-1}$) that it leads to a force that surpasses the force produced by the underlying free energy barrier $-\frac{1}{2}\omega_{eq}^2(s - s^{\ddagger})$, where $\omega_{eq}$ = 409cm$^{-1}$. This indicates that the motion of the system at the TS enters the so-called polarization caging regime (different friction regimes in GH theory are discussed in the SI)[42]. In this regime, the environment can dynamically trap the system in the TS region, so that relaxation of the system from the TS is controlled by the motion of the coupled degrees of freedom in the environment. The analysis of the free trajectories started at the TS confirms such a caging regime, where the system remains in the TS region for several ps in some of the trajectories (see Supplementary Fig. 5).

A unique feature of our GH approach is that it provides a detailed determination of the degrees of freedom responsible for the friction on loop motion. To determine which degrees of freedom are coupled to the reaction coordinate we recalculated the friction coming from the forces exerted by the solvent and the protein separately, taking advantage of the pairwise nature of the force field. We further decomposed the protein contribution to the friction into inter- and intramolecular contributions (backbone and sidechain torsions). Figure 4a shows the different contributions to the total friction including solvent and protein intramolecular contributions (stretching, bending, and torsions). This demonstrates that the protein term almost entirely determines the friction acting on the loop conformational change. Figure 4b shows the power spectrum of the friction, focusing on the lower frequency region. Slower movements are responsible for the deviation of the transmission coefficient from unity since these motions may lag behind the progress of the system along the reaction coordinate, causing the trajectory to return to the reactants valley or the caging effect mentioned above. An analysis of the friction power spectrum indicates that some stretching/bending contributions appear in the 200–700 cm$^{-1}$ region while torsions mainly appear in the region below 400 cm$^{-1}$ and constitute the only contribution below 200 cm$^{-1}$. Among these torsions, ϕ and ψ backbone angles make the most significant contribution to the friction, indicating the resistance of the loop backbone to follow the rotation of the Asp181-Phe182 peptide group. Supplementary Fig. 6 shows a Fourier Transform analysis of the motion of the ϕ and ψ backbone angles coupled to the reaction coordinate at the TS, which shows that the low-frequency contributions to the friction are largely dominated by two particular

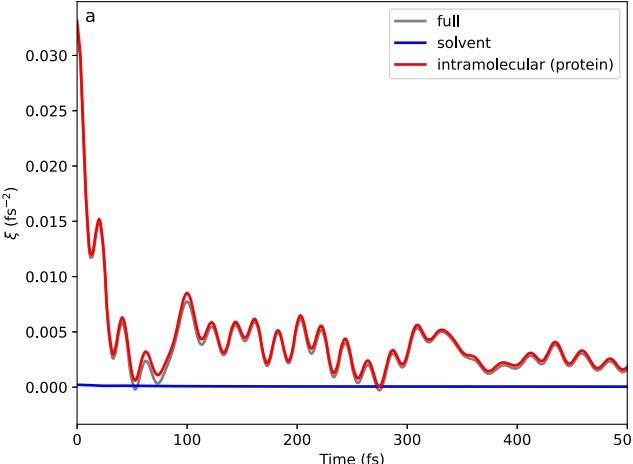

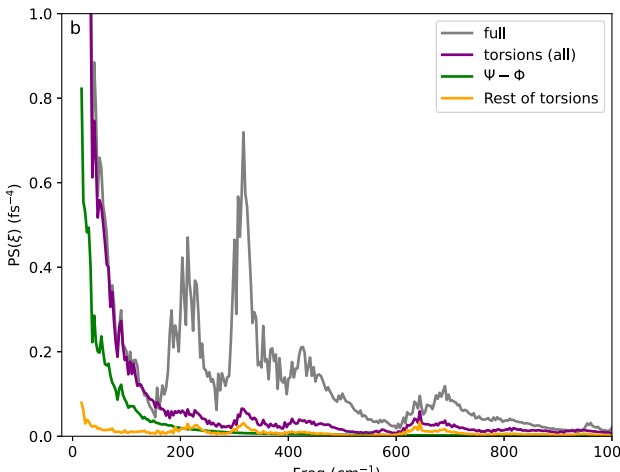

**Fig. 4 | Contributions to the friction acting on the conformational change of the WPD-loop in PTP1B. a** Time-dependent friction acting on the reaction coordinate for the closed-to-open loop transition in PTP1B, calculated at the TS. The total friction is decomposed into contributions from solvent and intramolecular

protein forces, which is responsible for almost all the friction; **b** Power spectra of the total friction and the contribution caused by torsions, separating the contributions of backbone torsions (ϕ and ψ) from the remaining protein torsions.

torsions, $\psi_{182}$ (from Phe182) and $\phi_{183}$ (from Gly183). As discussed below, this indicates that the magnitude of the torsional friction is sequence-dependent.

This study presents a rigorous decomposition of the contributions to the friction that occurs during a protein conformational change (within the pairwise approximation of the force field). Our findings reveal that the torsional motions of the loop backbone resist the conformational change. In the context of protein folding, time-resolved fluorescence anisotropy has been used to show that short-range backbone dihedrals cause the friction acting during conformational transitions of intrinsically disordered proteins[22]. Furthermore, simulations of peptide and protein folding processes have shown that internal friction effects can be ascribed to torsional barriers[19,41]. In the case of the loop opening and closing motion in PTP1B, we have demonstrated that barrier crossing takes place under strong internal friction due to the accommodation of those torsions that are coupled to the conformational change. This coupling can be efficiently captured as a time-dependent friction acting on a properly chosen reaction coordinate. It must be emphasized that while friction can account for a rate constant reduction roughly by a factor of 10, the barrier height remains the main factor controlling loop kinetics.

Our proposed mechanism for the opening/closing conformational change of a loop consists of two types of motions: diffusive displacement and activated torsional rearrangement. The torsional rearrangement occurs around a specific peptide group (Asp181-Phe182 in the WPD-loop of PTP1B). The resistance of the rest of the loop to the conformational transition can be considered as a frictional force acting on the reaction coordinate. We now demonstrate that this picture is general and we show how it applies to other protein loops, first for another member of the PTP family and then for a different type of enzymes.

## The YopH case

A particularly interesting system is the YopH, another member of the PTP family that also contains a WPD-loop, like PTP1B. The WPD-loop of YopH also has a β-turn but composed in this case by residues Pro355-Asp356-Gln357-Thr358. Another difference between the two enzymes is that the catalytic aspartate of YopH (Asp356) forms a salt bridge with an arginine (Arg409) only in the closed state (apo form), whereas in the open state, Asp356 interacts with Ser289. The main difference in conformation between the loop backbones in the open and closed forms of the YopH loop is the rotation of the peptide group Asp356-Gln357 (see Supplementary Fig. 2). Interestingly, the experimental rate constant for loop opening in YopH is about two orders of magnitude larger than that of PTP1B[25]. As explained in the "Methods" section, we followed a similar computational protocol to that used in PTP1B to estimate the rate and equilibrium constants for the open/closed conformational change of the WPD-loop in YopH. The values obtained for the rate and equilibrium constants for the loop change in YopH (see Table 1) are in very good agreement with the experimental observations.

To understand the similarities and differences between the conformational changes of the WPD-loop in PTP1B and YopH, we analyze the MFEP obtained for the latter in Fig. 5a. The picture obtained for the loop conformational change in YopH is very similar to that described for PTP1B: the process is a combination of a diffusive displacement of the loop and an activated rotation of the Asp356-Gln357 peptide group (see evolution of distances and torsional angles in Fig. 5b). After the free energy barrier, the open state valley presents a first minimum corresponding to the formation of a transient salt-bridge interaction between residues Asp356 and Lys447, with an average Cγ-Nζ distance of $3.6 \pm 0.3$ Å. Once this interaction is broken, the system evolves towards a completely open structure. The free energy barrier in YopH is about 3 kcal·mol⁻¹ smaller than in the case of PTP1B, which explains the observed increase in the rate constants. One contribution to the

reduced activation free energy for loop opening in YopH with respect to PTP1B is the composition of the β-turn. The residue in the fourth position of this β-turn is bulkier in YopH than in the WPD-loop of PTP1B (Thr358 in YopH versus Gly183 in PTP1B). This results in a weaker hydrogen bond between the carbonyl group of the first residue and the NH group of the fourth residue of the β-turn in the case of YopH, as seen in Supplementary Fig. 2. This difference may be the origin of the increased free energy barrier in PTP1B, because this intra-β-turn hydrogen bond must be broken at the TS (see Supplementary Fig. 7).

The time-dependent friction acting on the reaction coordinate for the loop change is presented in Fig. 5c. The friction is smaller for YopH than in the case of PTP1B, but still corresponds to a caging regime where the initial friction is larger than the equilibrium reaction frequency (see Supplementary Table 6). The transmission coefficient obtained for YopH is larger than for PTP1B: 0.24 versus 0.17, reflecting the smaller friction. This friction in YopH is also dominated by intra-molecular contributions to the force field, and the role of inter-molecular interactions is negligible. The power spectra (Fig. 5d) shows that the reduced friction in YopH compared to PTP1B is due to the smaller contribution of backbone torsions in YopH, which dominate the friction at low frequencies below 50 cm⁻¹, while in PTP1B backbone torsions were predominant already at 200 cm⁻¹. The comparison of the Fourier transforms of the time evolution of the backbone torsions coupled to the reaction coordinate show that the lack of Phe and Gly at positions three and four of the β-turn explains the differences observed in the friction acting on both enzymes These observations suggest that the internal friction arising due to torsional relaxation is controlled by the local sequence composition, in agreement with a recent experimental study on the origin of friction in the folding of intrinsically disordered proteins[43].

## Extension to other enzymatic systems

Our picture of the protein-loop opening mechanism applies to a broad range of systems beyond the PTP family. Upon inspection of several X-ray structures of enzymes with loops in open and closed conformations, we have observed that the torsional difference between the backbones in the two conformations is primarily due to a local rotation around a single peptide group. In the case of Triosephosphate Isomerase, the open and closed forms of loop 6[44,45] differ in the orientation of the carbonyl and amide groups of the Leu174-Ala175 peptide group. A similar difference is observed for the loop closing the active site in Lactate Dehydrogenase (Arg105-Leu106 peptide group in the rabbit version)[46]. Finally, in the case of the Zika virus helicase[47], the conformational change of motif V involves the rotation of the Met414-Gly415 peptide group[48].

A paradigmatic example for loop conformational changes is the M20 loop of *E. coli* Dihydrofolate Reductase (EcDHFR)[49]. This loop is key for catalysis and it experiences a conformational rearrangement from closed-to-open and then to occluded conformations along the catalytic cycle. The closed conformation of the M20 loop, found in the Michaelis complex, is stabilized by hydrogen bonding to the FG loop. After the chemical step, the M20 loop adopts the occluded conformation, stabilized by hydrogen bonds to the neighboring GH loop. NMR experiments indicate that the exchange between M20 conformations occurs in the millisecond timescale, being sensitive to the presence of different ligands[50]. We simulated here the closed-to-open part of the complete transition using a protocol similar to that applied to the members of the PTP family. For the selection of CVs, we ran MD simulations of the closed and open states and observed little or no overlap for the probability distributions of the $\psi_{17}$ and $\phi_{18}$ torsional angles that define the rotation of the peptide group Glu17-Asn18. We also observed no overlap for the probability distributions of the contiguous $\phi_{17}$, indicating a very strong coupling of this torsional angle with the rotation of the Glu17-Asn18 peptide group (see Supplementary Fig. 8) and then $\phi_{17}$ was also included as a CV. This finding agrees

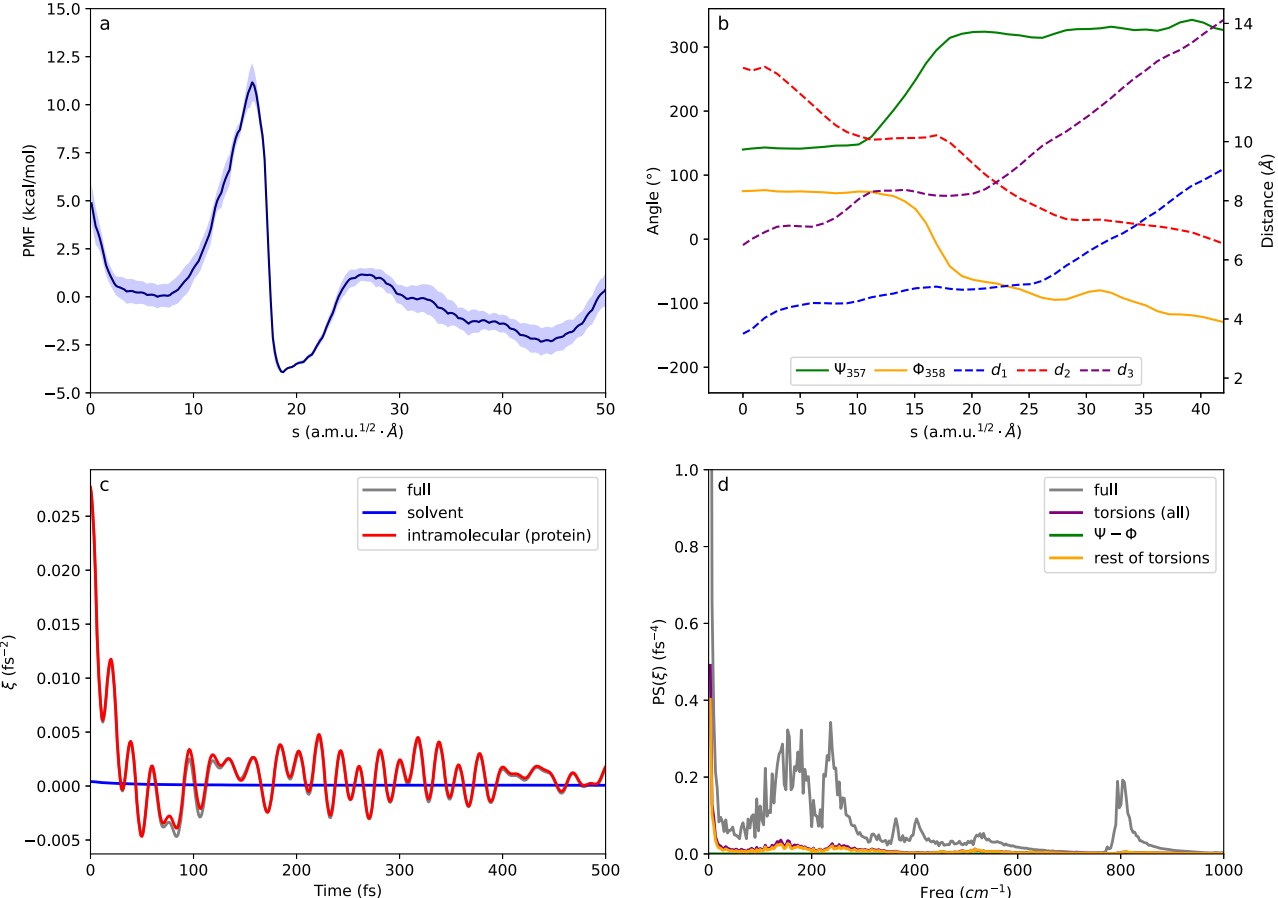

**Fig. 5 | Results for the closed-to-open conformational change of the WPD-loop in YopH. a** Free energy profile for the closed (left) to open (right) transition along the $s$ path-CV as obtained from the string method. The shaded region corresponds to the statistical uncertainty (95% CI); **b** Evolution of the individual CVs (distances in right vertical axis and dihedrals left vertical axis) along the MFEP. The selected CVs are the torsional angles $\psi_{357}$, $\phi_{358}$ and the distances Asp356Cγ-Arg409Cζ ($d_1$), Asp356Cγ-Ser289 Oγ ($d_2$) and Asp356Cα-Gly408Cα ($d_3$). **c** Time-dependent friction

acting on the reaction coordinate for the closed-to-open loop transition calculated at the TS. The total friction is decomposed in contributions coming from solvent and intramolecular protein contributions, the latter is responsible for most of the friction; **d** Power spectra of the total friction and the contribution due to torsions, with the contributions of backbone ($\phi$ and $\psi$) separated from the rest of the protein torsions.

with the friction analysis presented above that emphasizes the role of contiguous torsions during the rotation of a peptide group. The distances selected to guide the displacement of the loop were Asn18Cγ-His45Cα and Asn18Cγ-Glu120Cα, that increase/decrease significantly from the closed to the open state, respectively. The first distance corresponds to a contact formed in the closed state, while the distance with respect to Glu120 defines the positioning of the Met20 loop relative to the FG one (see Supplementary Fig. 8c). The results of the string calculations carried out for this set of 5 CVs is shown in Fig. 6. The conformational change from closed-to-open present an activation free energy of 6.5 kcal·mol⁻¹ (Fig. 6a), which is compatible with the values observed for the complete conformational transition[50]. The loop RMSD values determined with respect to the X-ray structures, show that our simulations successfully lead the protein from the closed-to-open state (Fig. 6b). As in the preceding examples the free energy profile shows two minima corresponding to the closed and open states separated by torsional barriers. The main contribution to this barrier is the rotation of the peptide Glu17-Asn18, as observed from the evolution of the CVs (Fig. 6c). In this case, the conformational readjustment of the contiguous torsion $\phi_{17}$ causes a second free energy barrier before $\psi_{17}$ and $\phi_{18}$ can relax to the values corresponding to the open state. Note that this second torsional barrier can be seen as an extreme case of strong friction leading to a caging regime discussed above, reflected here in the existence of a high-energy

intermediate. MD simulations (see Supplementary Fig. 9) show that this intermediate can exist during tens of ns before relaxing to the open state. The results obtained in the case of the EcDHFR M20 loop thus confirm the picture derived from the simulations of PTPs enzymes: loop conformational transitions can be described as a local change in a peptide group coupled to neighbor backbone torsions.

## Discussion

Loop motions are crucial constituents of protein dynamics during the catalytic cycle of many enzymes. Loop opening and closing allow the binding of the substrate in the active site and/or bring different reaction partners together within adequate distances and orientation. Therefore, understanding the mechanisms that govern loop motions is necessary to rationalize enzyme behavior and to engineer better biocatalysts. In this study, we conducted a computational analysis of the opening/closing conformational change of the WPD-loop in two PTP enzymes, PTP1B and YopH, and of the M20 loop in EcDHFR. The WPD-loop loop contains one of the key residues for the phosphatase activity of these enzymes, Asp181 or Asp356, and loop closing over the active site is a necessary step for catalysis. Our simulations are based on a path-collective variable that depends on a combination of few distances and torsions that define the position of the loop and the conformational changes in the backbone. The rate constants for the loop opening and closing transitions obtained within this picture are in

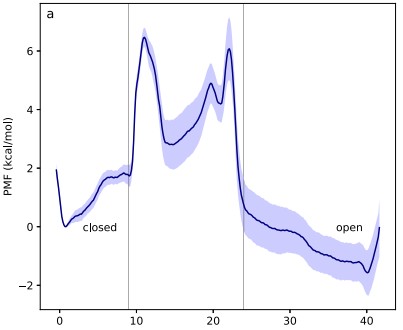
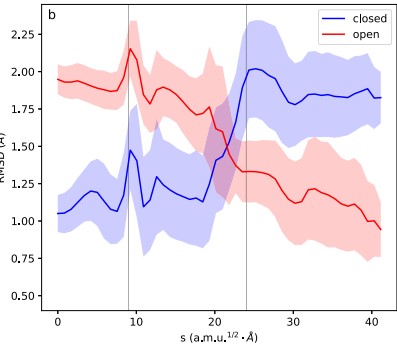
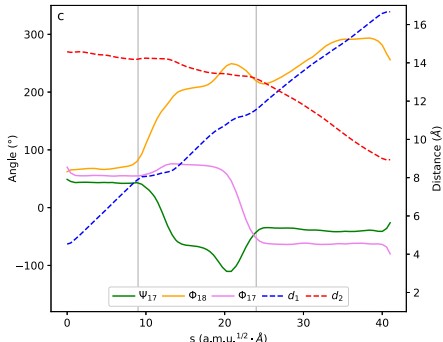

**Fig. 6 | Free energy landscape for the M20 loop closed-to-open conformational change in EcDHFR. a** Free energy profile for the closed (left) to open (right) transition along the *s* path-CV. The shaded region corresponds to the statistical uncertainty (95% CI); **b** Average RMSD measured for the backbone atoms of the M20 loop for snapshots obtained from the Umbrella Sampling simulations along the path-CV with respect to the X-ray structures (PDB 1RA1 and 1RX2) corresponding to the closed (blue) and open (red) states. The shaded region corresponds to the statistical uncertainty; **c** Evolution of the individual CVs (distances on the right vertical axis and dihedrals on the left vertical axis) along the MFEP. The CVs used in the ASM calculations are: $\phi_{17}$, $\psi_{17}$, $\phi_{18}$ torsional angles and the distances Asn18Cγ-His45Cα ($d_1$) and Asn18Cγ-Glu120Cα ($d_2$).

excellent agreement with experimental observations for both the WPD-loop of PTP1B and YopH. According to our findings, the transition of the loop between the closed and open conformations can be described as a combination of a diffusive displacement of the loop and a local torsional rearrangement that requires crossing a significant barrier. This free energy barrier is associated with the torsional rearrangement of the two dihedral angles that govern the rotation of a single peptide group, Asp181-Phe182 or Asp356-Gln357 in PTP1B and YopH, respectively. The same general picture holds in the case of the closed-to-open transition for the M20 loop of EcDHFR, where the rotation of the Glu17-Asn18 determines the conformational transition.

The conformational change defined by a ψ/φ pair has been shown to be strongly coupled to the rest of loop backbone torsions and, in particular, with the contiguous torsional angles. The loop backbone must be adapted to the local conformational change around the peptide group, offering a resistance that can be described as a strong friction acting during the barrier crossing event, as in the case of PTP enzymes, or as an additional contribution to the free energy barrier, as in the case of EcDHFR. The coupling of other degrees of freedom, particularly those of the solvent, is of minor importance here. We have also shown that the friction is correctly incorporated to the rate constant using Grote–Hynes equation, while Kramers' approximation overestimates the effect of this friction by ignoring its frequency-dependence. For the loop conformational transition, the friction is so strong that the barrier crossing is found in the so-called polarization cage regime. In this regime, the motion of the slow environmental degrees of freedom, such as the rest of torsions of the loop backbone, is required for the system to relax off the transition state region. In the case of EcDHFR the caging effect is so intense that it manifests itself with the presence of an intermediate when an additional torsional angle is incorporated in the reaction coordinate. These observations are also relevant for the treatment of internal friction in the study of protein folding processes, which is mainly due to torsional degrees of freedom.

Engineering of enzymatic loops has attracted an increased attention as a strategy to alter enzymatic function, stability, and specificity. The framework presented here offers an opportunity to rationalize the consequences of mutations on loop kinetics. It reveals how mutations can increase/decrease loop kinetics by decreasing/increasing the torsional barrier associated to the backbone rearrangement. This is mostly a local effect around a particular peptide group that can be understood in terms of changes in protein-loop and/or intra-loop interactions. On the other side, mutations can also contribute to a fine-tuning of loop kinetics through a change in the friction, which is a more collective effect involving the rearrangement of the whole loop backbone. This picture can be extended not only to loop motions in other enzymatic systems but also to lid motions and conformational changes of larger protein motifs[51], which could involve a few key torsional rearrangements determining the barrier height accompanied by an extensive backbone readaptation entering as a friction into the rate constant. As the importance of protein conformational dynamics is increasingly recognized, identifying the molecular rearrangements that control its kinetics is critical.

## Methods

### Preparation of the systems

The simulation PTP1B system was based on the PDB structure 6B90, which contains the enzyme with the studied loop in both states (closed and open). The missing residues were taken from PDB structure 4Y14 after aligning it using PyMOL[52]. The hydrogen atoms and tautomeric states were assigned with pdb4amber from AmberTools20. The protonation states were assigned with Propka 3.0[53] at pH = 7.0. The protonated PTP1B structures (with closed and open loop) were described using the ff19SB force field[54]. OPC water[55] box and sodium counterions were added with tleap from AmberTools20[56]. The two systems were then minimized, the simulation box size was relaxed by running 100 ps of NPT MD followed by 100 ns of NVT MD using pmemd of Amber20[56]. Full details of the equilibration and simulation protocol are given in the SI. Except when indicated, all simulations employed a timestep of 2 fs and the SHAKE algorithm[57] was used to constrain bond lengths. Periodic boundary conditions were applied, long-range electrostatic interactions were treated with PME[58] and a non-bonded cutoff distance of 8.0 Å was used for van der Waals interactions. Langevin thermostat was used to control the temperature with a collision frequency of 2.0 ps⁻¹. The same system preparation and relaxation protocol was used for the YopH system. In this case the starting PDB structures for the closed and open states were 1YPT and 2I42, respectively, removing the vanadate ion present in the active site of the latter. Missing C-terminal residues in 1YPT were added from 2I42. In the case of EcDHFR the reference X-ray structures for the closed and open states were 1RX2 and 1RA1, respectively. To simulate the apo form, substrate and cofactor were removed.

See "Data availability" section for the files containing parameters and relaxed structures of the enzymes in the two states.

### Conventional molecular dynamics

The thermodynamic ensembles with the loop in closed and open states were obtained by running 10 × 100 ns NVT simulations in each

of the two states, starting from snapshots taken every 5 ns from the last 50 ns of the relaxation MD. To probe the response of the loop to the conformational change in the β-turn of PTP1B, 20 replicas starting from different structures with "closed" loop were run with dihedral restraints that force the flip of the peptide bond. Then the restraints were removed, and the replicas were allowed to evolve for 1 μs each. These MD simulations used the same specifications as indicated above.

### String simulations

Adaptive string method (ASM)[36,59] is an equilibrium approach to localize the minimum free energy path between two minima. Here ASM calculations were performed to capture both the local conformational change (controlled by torsional angles) and the loop displacement (controlled by distances). In ASM simulations are carried out over a series of replicas of the system (string nodes) centered at different positions in the space of collective variables (CVs) formed by the set of dihedrals and distances. The string nodes (see SI for details) evolve towards lower free energy regions while being evenly distributed, which ensures convergence to the MFEP. Half of the nodes were initiated with different uncorrelated structures taken from the simulations of the closed state and half from different structures of the open state. Hamiltonian replica exchange was attempted between neighboring string nodes every 250 simulation steps, improving convergence. Once the string has converged, a single path-CV, denoted as $s$, is used as a reaction coordinate for subsequent umbrella sampling free energy calculations, accumulating 10–15 ns per node. The initial guesses and the definitions of CVs for both string calculations are provided in the SI together with details about convergence and production times to obtain the free energy profiles.

### Rare event simulations

60 structures were selected from the TS simulation window, with values of the path-CV differing less than ±0.1 a.m.u.$^{1/2}$·Å from the value corresponding to the maximum of the free energy profile. Trajectories were initiated assigning velocities taken from a Maxwell-Boltzmann distribution. Trajectories were propagated forward ($t > 0$) and backward ($t < 0$) in time reversing the sign of initial velocities. Trajectories were propagated during 2x100 ps with a timestep of 2 fs. The outcome of the trajectories was classified as open/closed state according to the values obtained for $\psi_{181}$, $\phi_{182}$ torsional angles.

### Calculation of equilibrium and rate constants and the associated free energies

The equilibrium constant between the open and closed forms of the loop were obtained after integration of the PMF along the reaction coordinate ($s$):

$$K_{eq} = \frac{\int_{s > s^{\ddagger}} C_s^{-1} \cdot e^{-\frac{G(s)}{k_B T}} ds}{\int_{s < s^{\ddagger}} C_s^{-1} \cdot e^{-\frac{G(s)}{k_B T}} ds} \tag{3}$$

where $C_s$ is a normalization constant with units of the $s$ coordinate (1 a.m.u.$^{1/2}$·Å) and $s^{\ddagger}$ is the position of the TS.

The TST rate constant can be obtained from equilibrium flux across the dividing surface defined by the path-collective variable $s$[39]:

$$k^{TST} = \frac{1}{2} \langle |\dot{s}| \rangle_{\ddagger} \cdot C_s^{-1} \cdot \exp\left[-\frac{\Delta G_s^{\ddagger}}{k_B T}\right] \tag{4}$$

The preexponential term in Eq. (4) contains the normalization constant and the average modulus of the velocity along the reaction coordinate at the TS that was obtained assuming a Maxwell-Boltzmann distribution and the reduced masses of the reaction coordinate at the

TS (0.917 and 1.032 for PTP1B and YopH, respectively). The term in the exponential contains the free energy difference between reactants (closed/open states) and the TS:

$$\Delta G_s^{\ddagger} = G(s^{\ddagger}) - k_B T \cdot \ln \int_s C_s^{-1} \cdot e^{-\frac{G(s)}{k_B T}} ds \tag{5}$$

The reaction and activation free energies were then derived from the equilibrium and rate constants, respectively:

$$\Delta G_{eq} = -k_B T \cdot \ln K_{eq} \tag{6}$$

$$\Delta G_i^{\ddagger} = -k_B T \cdot \ln \frac{k_i h}{k_B T} = -k_B T \cdot \ln \frac{\kappa k_i^{TST} h}{k_B T} \tag{7}$$

where $i$ stands for closing/opening processes. The contribution of the transmission coefficient to the activation free energy can be evaluated as:

$$\Delta G_{\kappa}^{\ddagger} = -k_B T \cdot \ln \kappa \tag{8}$$

### Grote−Hynes simulations

Using the fluctuation-dissipation theorem, the time-dependent friction acting on the reaction coordinate at the TS can be calculated from the autocorrelation of the forces projected on the reaction coordinate ($F_s$):

$$\xi(t) = \frac{1}{k_B T} \langle F_s(0) \cdot F_s(t) \rangle_{\ddagger} \tag{9}$$

where we assumed mass-weighted coordinates.

Under the effect of this friction, the reaction frequency ($\omega_r$) for crossing a free energy barrier equal to $-\frac{1}{2}\omega_{eq}^2(s - s^{\ddagger})$ is given by the GH equation[33]:

$$\omega_r^2 - \omega_{eq}^2 + \omega_r \cdot \int_0^t \xi(t) \cdot e^{-\omega_r \cdot t} \cdot dt = 0 \tag{10}$$

The difference with respect to Kramers' theory is due to the fact that the effect of the friction is modulated by the reaction frequency, appearing inside the integral in Eq. (10). Using GH equation, the transmission coefficient appearing in Eq. (2) is simply obtained as the ratio between both frequencies:

$$\kappa = \frac{\omega_r}{\omega_{eq}} \tag{11}$$

The friction was obtained as an average over 60 independent TS trajectories, where the initial configurations were selected from string simulations at the top of the PMF. To decompose the different contributions to the friction, the forces were recalculated using the same configurations but zeroing different contributions to the MM force field. Details of Grote−Hynes simulations are provided in the SI.

### Reporting summary

Further information on research design is available in the Nature Portfolio Reporting Summary linked to this article.

## Data availability

The raw data, notebooks for figures, parameters files and coordinates used in this study are available in the Zenodo database under accession code https://doi.org/10.5281/zenodo.10670397.

PDB data: 6B90, 4Y14, 1YPT, 2I42, 1RX2 and 1RA1.

## Code availability

The adaptive string method is available in the Zenodo database under accession code https://doi.org/10.5281/zenodo.10670371 and in GitHub https://github.com/kzinovjev/string-amber.

Codes for data analysis and representation are available in the Zenodo database under accession code https://doi.org/10.5281/zenodo.10670397.

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

## Acknowledgements
PID2021-123332OB-C22 funded by MCIN/AEI/10.13039/501100011033/ and by "ERDF A way of making Europe" K.Z., J.J.R-P and I.T. PROMETEO CIPROM/2021/079 of Generalitat Valenciana. C.A.R-G, J.J. R-P and I.T. Maria Zambrano fellowship by Ministerio de Universidades (Spain). K.Z. 'Salvador de Madariaga' grant of MCIN (Spain). I.T. acknowledges the warm hospitality of the Département de Chimie (École Normale Supérieure, Paris).

## Author contributions
K.Z., D.L., J.J R-P., and I.T. were responsible of the design of the work. K.Z., P.G., and C.A. R-G. performed the calculations and all the authors participated in the discussion and analysis. K.Z. wrote the code needed to use Grote–Hynes theory and the string method. I.T., K.Z., and D.L. wrote the draft, while all the authors participated in the revision and editing of the manuscript.

## Competing interests
The authors declare no competing interests.
