## [Peer Review File · Nature Communications]

Activation and friction in enzymatic loop opening and closing dynamicsReviewer #1 (Remarks to the Author):

This study on loop dynamics by Zinovjev et al. combined MD simulations and a string-method approach to obtain the free energy profile and identified the transition state for the opening process of WPD-loop from PTP1B. They further dissect the loop opening process to include a diffusive displacement of the loop, and more importantly, a torsional rearrangement at the loop backbone. The free energy barrier for the opening process is largely associated with the torsional rearrangement. The authors went on to show that this finding could be a more general scenario, with similar trends observed in another member of the PTP family that also contains a WPD-loop, TopH. While the findings may be of interest, the manuscript is highly complex and as a result, some areas were not written clearly and provided insufficient detail. Several main concerns:

1. While it is not surprising that loop movement involves backbone torsional changes, the authors' claim on "general applicability" in this study is not fully supported by the evidence provided in the manuscript. Both examples were members of the same enzyme family with the same loop motion. For "general" applicability, I would expect to see an example of this on an unrelated family of enzymes.

2. The calculations to obtain the free energy profile for loop opening was based on single starting and end points. That is, starting from one closed loop conformation, and with the string method defined reaction coordinate, using umbrella sampling to drive the system to a single end conformation, i.e. the open state. This would be sufficient if it is known that both open and closed states sample only a single energy minimum. However, as the authors have also noted, there is a wide range of structures for the open state, with different conformations. The free MD simulations also showed diverse range in the salt bridge distances in the open state. Would the free energy profile and the transition process change if a different open conformation was selected as the end point? At least, a thorough examination on all the open state structures to collect the relevant parameters, including the backbone torsions, and the two distances d_1 , d_2 , need to be done, to justify using this particular structure in the open loop as the end point.

3. In several places of the manuscript, the authors claimed "excellent" agreement between calculation and experiment. However, it is very difficult to see how the numbers in Table 1 can be taken as showing agreement between calculation and experiment. For PTP1B, the calculated k_{opening} and k_{closing} not only failed to predict values close to those measured by experiment, they also don't observe the same trend, i.e. k_{closing} being much smaller than k_{opening} . There are similar concerns for the calculations for YoPH. One order of magnitude difference is hardly "excellent". Page 12, line 359, the authors commented: "the value predicted by GH theory agrees with the calculated ratio of reactive trajectories estimated from the free trajectories initiated at the TS". The calculated value from GH theory is 0.17, the ratio of reactive trajectories is 0.22 (13 out of 60), they obviously don't agree exactly however there is no reference of standards to indicate how far the two values have to be before they "don't agree".

4. In figure 5a, there is a low energy minimum after the transition state, then moving on from this energy minimum towards the final open state structure, the system goes through a second energy barrier. What is causing that second energy barrier and why TopH's open state energy profile is not as flat as PTP1B?

5. The authors studied the process of loop opening, i.e. starting from a closed state and move to an open state. A relevant scenario is the transition from open to close state, as this is the motion that will deliver catalysis. Why did the authors choose to study the reverse process? What would be different in the free energy profile of the transition is computed from the open state to the closed state?

Other minor concerns:

1. Figure 2b legend should be for the "closed" conformation, not "open" conformation.

2. Page 8, line 241-244, "Note that according to our simulations, the free energy of the open state lowers because the corresponding valley along the reaction coordinate is significantly wider and thus that state contains a larger number of possible structures, while the free energy profile itself favors the closed state." This sentence doesn't make sense. A wider energy minimum with larger number of possible conformations is not necessarily lower, in fact the free energy profile is showing exactly what the free MD is suggesting, that the open state has a wide energy minimum. However, the free MD simulations do not suggest the open state is lower in energy.

3. It is very hard to understand figure 3 on initial reading of the manuscript, please label on the

figure clearly where are the “three stages”, and label open and close states clearly.

4. Page 10, line 287-290, “free molecular dynamics simulations for closed state configurations, flipping only the peptide bond...”, this sentence is confusing as free MD simulations implies no external forces, and flipping peptide bond indicated external forces have been applied.

5. Table S5, please give clear detail how these values have been derived, especially the three values with cm^{-1} units, the origin of these number values have not been explained or referenced anywhere in the manuscript.

6. Figure S5. Please explain what “positive time/negative time” is, it doesn’t make sense in terms of MD simulations. Please also explain how the trajectories are classified as “reactive”, and authors should clearly label or indicate which trajectories in Figure S5 is reactive so readers can see clearly.

Reviewer #2 (Remarks to the Author):

Zinovjev et al present a well-structured study on the topic of loop opening/closing dynamics in the PTP1B enzyme. They carried detailed analysis on the WPD-loop dynamics with carefully chosen collective variables. Strong evidence is provided to demonstrate the possible mechanism of loop displacement and loop conformational rearrangement. Kinetic study is also carried out to make the study comparable to experimental measurements. However, some limitations still exist, making the report difficulty to understand.

1. In Figure 2 a, some probability distributions indicate the presence of different modes. How do authors ensure their simulations are long enough to represent the converged distributions? In addition, they carried out 10 individual MD with each 100 ns. Is the MD setup sufficiently converged in sampling the intended distributions?

2. In Figure 3 a, the open state basin seems to have more uncertainties and is explained as such that the open state allows more conformations and thus more stable. Can authors show additional evidence on the stabilities of the two states? Figure 3b seems to present the RMSD of backbone atoms of WPD-loop along the path-CV. Do they use the averaged RMSD at each simulation with individual CV values? Why is the open state path monotonic compared with the closed state path?

3. In the kinetic study, the computed forward and backward rate constants turn out to be the same for PTP1B. Is this because the computed k^{TST} constants are the same for the forward and backward, given that the transmission coefficient is the same? The qualitative agreement with experiment appears to be better for YoPH than that for PTP1B. Also, what are the uncertainties for the computed rates? It is good to know to access the agreement.

In the following pages, we provide detailed answers to all the points raised by the reviewers and explain the changes introduced in the revised version.

Answers:

Reviewer #1

This study on loop dynamics by Zinovjev et al. combined MD simulations and a string-method approach to obtain the free energy profile and identified the transition state for the opening process of WPD-loop from PTP1B. They further dissect the loop opening process to include a diffusive displacement of the loop, and more importantly, a torsional rearrangement at the loop backbone. The free energy barrier for the opening process is largely associated with the torsional rearrangement. The authors went on to show that this finding could be a more general scenario, with similar trends observed in another member of the PTP family that also contains a WPD-loop, TopH. While the findings may be of interest, the manuscript is highly complex and as a result, some areas were not written clearly and provided insufficient detail. Several main concerns:

We would like to thank the reviewer for his/her positive reaction to our work. In the following we address the reviewer's concerns and detail the extensive changes made to clarify our approach and to make it accessible to the widest audience.

1. While it is not surprising that loop movement involves backbone torsional changes, the authors' claim on "general applicability" in this study is not fully supported by the evidence provided in the manuscript. Both examples were members of the same enzyme family with the same loop motion. For "general" applicability, I would expect to see an example of this on an unrelated family of enzymes.

Following the reviewer's advice, we have added a new system to our manuscript and performed extensive additional simulations. We selected for this additional study a paradigmatic example for loop conformational changes, the M20 loop of *E. coli* Dihydrofolate Reductase. This loop is key for catalysis and undergoes a conformational rearrangement from closed to open and then to occluded conformations along the catalytic cycle. The closed conformation of the M20 loop, found in the Michaelis complex, is stabilized by hydrogen bonding to the FG loop. After the chemical step, the M20 loop adopts the occluded conformation, stabilized by hydrogen bonds to the neighboring GH loop. We simulated the closed to open transition using the same protocol as that applied to the members of the PTP family. Our results show that the conformational change can be attributed mostly to the rotation of the Glu17-Asn18 peptide group, strongly coupled to neighbor torsions.

We think that with this new example added to the manuscript, the mechanism proposed for loop motion, a combination of displacement and a local conformational rearrangement around a peptide group strongly coupled to other loop backbone torsions, can be considered as a plausible mechanism for a broad range of cases, clearly extending the applicability and interest of our proposal. Our combined results on two PTP proteins and DHFR demonstrate the general applicability of our simulation approach and of our proposed mechanism for protein loop motion.

2. The calculations to obtain the free energy profile for loop opening was based on single starting and end points. That is, starting from one closed loop conformation, and with the string

method defined reaction coordinate, using umbrella sampling to drive the system to a single end conformation, i.e. the open state. This would be sufficient if it is known that both open and closed states sample only a single energy minimum. However, as the authors have also noted, there is a wide range of structures for the open state, with different conformations. The free MD simulations also showed diverse range in the salt bridge distances in the open state. Would the free energy profile and the transition process change if a different open conformation was selected as the end point? At least, a thorough examination on all the open state structures to collect the relevant parameters, including the backbone torsions, and the two distances d_1 , d_2 , need to be done, to justify using this particular structure in the open loop as the end point.

The reviewer rightfully points out the importance of adequate sampling in the starting and end conformations. This was explicitly considered in our study, but our presentation was evidently not clear enough. In contrast to the impression that the Reviewer had, our simulations did not consider transitions between a single open structure and a single closed structure. Instead, in our string simulations each of the 60 nodes was started from a different uncorrelated structure. For the first 30 nodes we chose 30 different closed structures and for the second half we took 30 different open structures, all of them obtained from long MD simulations. This is, during the string method simulations each node was started from a different initial structure. In addition, during the string simulations, replica exchanges (attempted every 250 simulation steps) enhanced the conformational sampling and propagated the favourable structural changes along the set of string nodes. This extensive sampling, of the initial and final states and along the transition pathway, ensures that our results do not depend on our choice of initial conditions and account for the protein conformational flexibility. We are therefore confident that our results provide the converged minimum free energy profile for the loop opening/closing motion. In the revised version we have added further details to the Method section included (String simulations) to clarify these points.

3. In several places of the manuscript, the authors claimed “excellent” agreement between calculation and experiment. However, it is very difficult to see how the numbers in Table 1 can be taken as showing agreement between calculation and experiment. For PTP1B, the calculated k_{opening} and k_{closing} not only failed to predict values close to those measured by experiment, they also don't observe the same trend, i.e. k_{closing} being much smaller than k_{opening} . There are similar concerns for the calculations for YoPH. One order of magnitude difference is hardly “excellent”. Page 12, line 359, the authors commented: “the value predicted by GH theory agrees with the calculated ratio of reactive trajectories estimated from the free trajectories initiated at the TS”. The calculated value from GH theory is 0.17, the ratio of reactive trajectories is 0.22 (13 out of 60), they obviously don't agree exactly however there is no reference of standards to indicate how far the two values have to be before they “don't agree”.

We thank the reviewer for calling our attention to this point. When we were talking about ‘excellent’ agreement we were considering the agreement in terms of free energies. Our activation free energies and ‘reaction’ free energies agree with the experimental derived values within 2 kcal/mol in all cases. Moreover, our results reproduce the experimental trends of the changes in loop dynamics between systems. Considering the approximate nature of the underlying potential energy function and limitations in the sampling we consider this agreement as being very good. Obviously, when this free energy difference is translated into a change in rate constant, differences can exceed one order of magnitude, because of the

exponential dependency, as well known in numerical studies of reaction kinetics. In the revised version of the manuscript, we have clarified this point and also toned down our claim about the agreement. We also added the statistical errors of the activation free energies in the new version of the manuscript.

Regarding the calculation of the transmission coefficient, it must be clarified that an estimation based on the ratio between reactive and non-reactive trajectories is only qualitative. The correct definition of the transmission coefficient is based on the flux, not the number of trajectories. Transmission coefficients calculated using Grote-Hynes theory have already been shown to be in excellent agreement with those obtained from MD simulations (see, for example, *J. Chem. Phys.* 1989, 90, 3537–3558; *J. Am. Chem. Soc.* 2006, 128, 6186–6193 or *J. Am. Chem. Soc.* 2008, 130, 7477–7488). Here we used that qualitative estimation only with the purpose of showing that Grote-Hynes estimation is in the expected order of magnitude while Kramers' approach overestimates the impact of the friction. We have clarified this point in the revised manuscript.

4. In figure 5a, there is a low energy minimum after the transition state, then moving on from this energy minimum towards the final open state structure, the system goes through a second energy barrier. What is causing that second energy barrier and why TopH's open state energy profile is not as flat as PTP1B?

Free energy landscapes associated to loop conformational changes may contain several shallow free energy minima (see, for example, *J. Am. Chem. Soc.* 2018, 140, 46, 15889–1590). In this sense, it is preferable to talk about open and closed valleys, where different minima may coexist. In the revised version we have characterized the first minimum that appears along the minimum free energy path in the closed to open transition in YopH. This is due to the formation of a transient salt bridge between residues Asp356 and Lys447.

5. The authors studied the process of loop opening, i.e. starting from a closed state and move to an open state. A relevant scenario is the transition from open to close state, as this is the motion that will deliver catalysis. Why did the authors choose to study the reverse process? What would be different in the free energy profile of the transition is computed from the open state to the closed state?

The string method is an equilibrium approach to localize the minimum free energy path between two states and then the path should not depend on the direction being considered. In fact, there is nothing in the method that presupposes the direction of the transition. The minimum free energy path would be the same for the open to close than for the close to open transitions. We have clarified this point in the revised manuscript (Methods).

Other minor concerns:

1. Figure 2b legend should be for the “closed” conformation, not “open” conformation.

We thank the reviewer for detecting this error. We have modified the figure caption to correct the error.

2. Page 8, line 241-244, “Note that according to our simulations, the free energy of the open state lowers because the corresponding valley along the reaction coordinate is significantly wider and thus that state contains a larger number of possible structures, while the free energy profile itself favors the closed state.” This sentence doesn’t make sense. A wider energy minimum with larger number of possible conformations is not necessarily lower, in fact the free energy profile is showing exactly what the free MD is suggesting, that the open state has a wide energy minimum. However, the free MD simulations do not suggest the open state is lower in energy.

We are sorry for the confusion caused by our sentence. The free energy profile $G(s)$ along the coordinate s is determined by the probability of each specific value of the coordinate s . In contrast, the relative free energies of the open and closed states are determined by the **sums** of the probabilities over the range of s values corresponding to the definition of each state. The equilibrium constant between open and closed states thus depends on the integrated probabilities $\exp(-G(s)/kT)$ over the s ranges defining each state. With our reaction coordinate s , the range of s values defining the open state is much broader than that for the closed state. This can be seen as an entropic contribution to the free energy, because the closed state is more constrained than the open state. We have changed the sentence mentioned by the reviewer to make this point clearer.

3. It is very hard to understand figure 3 on initial reading of the manuscript, please label on the figure clearly where are the “three stages”, and label open and close states clearly.

We have changed figure 3 to introduce the changes suggested by the reviewer.

4. Page 10, line 287-290, “free molecular dynamics simulations for closed state configurations, flipping only the peptide bond...”, this sentence is confusing as free MD simulations implies no external forces, and flipping peptide bond indicated external forces have been applied.

We have changed the sentence pointed out by the reviewer to clarify that the initial structures were prepared out of equilibrium by flipping the peptide group (not the peptide bond) for configurations taken from the closed state. Then, starting from these configurations, and without further bias, free MD simulations were run, observing a diffusive behaviour for loop opening.

5. Table S5, please give clear detail how these values have been derived, especially the three values with cm^{-1} units, the origin of these number values have not been explained or referenced anywhere in the manuscript.

We have now added explicitly in Table S5 captions (Table S6 in the new version) the reference to the equations that connect frequencies with friction (see equations S1 to S5 in SI).

6. Figure S5. Please explain what “positive time/negative time” is, it doesn’t make sense in terms of MD simulations. Please also explain how the trajectories are classified as “reactive”,

and authors should clearly label or indicate which trajectories in Figure S5 is reactive so readers can see clearly.

Rare event simulations are started from selected TS configurations and propagated forwards and backwards in time just reversing the sign of the assigned velocities, allowing the reconstruction of full trajectories that starting at reactants/products reaches the TS and then evolves to any of the two possible valleys (see J. Am. Chem. Soc. 1991, 113, 74-87). By convention the time origin is taken to be the moment when the trajectory reaches the TS configuration; times before the TS crossing are defined as negative, and times after the crossing are positive. We have added a subsection about how rare events simulations have been carried out in the Methodological section of the manuscript.

Reviewer #2:

Zinovjev et al present a well-structured study on the topic of loop opening/closing dynamics in the PTP1B enzyme. They carried detailed analysis on the WPD-loop dynamics with carefully chosen collective variables. Strong evidence is provided to demonstrate the possible mechanism of loop displacement and loop conformational rearrangement. Kinetic study is also carried out to make the study comparable to experimental measurements. However, some limitations still exist, making the report difficulty to understand.

We thank the reviewer for their positive evaluation of our work, emphasizing the insight provided by our coordinates. As detailed below, in our revision we address a number of points to clarify our method and its generality.

1. In Figure 2 a, some probability distributions indicate the presence of different modes. How do authors ensure their simulations are long enough to represent the converged distributions? In addition, they carried out 10 individual MD with each 100 ns. Is the MD setup sufficiently converged in sampling the intended distributions?

The purpose of these relatively short simulations was to explore different configurations in the open and closed valleys and to identify good candidates to be used as collective variables for the string simulations. Running several independent shorter simulations with different starting configurations can be a better strategy for this purpose than running only one or a few longer MD simulations (see for example J. Chem. Inf and Model. 2016, 56, 1950-1962). Once the CVs and different initial configurations were selected, we run the string method for PTP1B for a total of 2.7 microseconds. During the string simulation we employed replica exchange to improve the convergence, propagating possible conformational changes. The statistical analysis of the error in the PMF profile shows that we have sufficiently explored the configurational space and that the profile is properly converged. In addition, 20 trajectories of 1 microsecond each were also run to explore the diffusion of the loop once the barrier was crossed. In these simulations we did not observe outliers to the distributions presented in Figure 2. The consistency between long free trajectories and enhanced sampling simulations allows us to be confident that our results are converged.

The methodological options have been now better described in the revised version of the manuscript.

2. In Figure 3 a, the open state basin seems to have more uncertainties and is explained as such that the open state allows more conformations and thus more stable. Can authors show additional evidence on the stabilities of the two states? Figure 3b seems to present the RMSD of backbone atoms of WPD-loop along the path-CV. Do they use the averaged RMSD at each simulation with individual CV values? Why is the open state path monotonic compared with the closed state path?

In our study we show that a correct definition of closed/open states must include the torsional angles that define the rotation of the 181-182 peptide bond. In none of our simulations at 300 K we observe a spontaneous rotation of the peptide bond and the two states are perfectly separated. When we forced this rotation for 20 configurations of the closed state the change is never reversed, which agrees with the significant free energy barrier observed in our free energy profile, and the trajectories show the diffusion of the loop towards an open configuration like that observed in the x-ray structure. We have now added a comment on these observations in the revised version of the manuscript.

The RMSD values reported in Figure 3 are averaged values obtained for snapshots obtained from the Umbrella Sampling simulation along the path-CV s . This has been now clarified in the caption of the figure. The non-monotonic behaviour observed when comparing to the closed structure is probably due to the perturbation caused by breaking the salt-bridge between Asp181 and Arg221. However, the statistical uncertainties associated to these values are large to draw any conclusion.

3. In the kinetic study, the computed forward and backward rate constants turn out to be the same for PTP1B. Is this because the computed $k^{\ddagger}TST$ constants are the same for the forward and backward, given that the transmission coefficient is the same? The qualitative agreement with experiment appears to be better for YoPH than that for PTP1B. Also, what are the uncertainties for the computed rates? It is good to know to access the agreement.

Our calculation of the equilibrium constant between the open and closed states was made using equation (1) that integrates the free energy profile for all the values of the coordinate compatible with the open or closed states. For the calculations of the rate constant, we used equation (4) and (5) where the free energy profile is also integrated for the reactants valley. The fact that the calculated rate constants are equal (or that the equilibrium constant is unity) results from the combination of two opposite trends: the lower free energy profiles values obtained for the closed state and a wider configurational space available for the open state. The fact that the configurational space available for the open state is wider also contributes to increase the uncertainty in the determination of its free energy. We added a comment on this observation in page 9 of the revised manuscript.

We have also added the statistical uncertainties associated to the activation free energies, which are of 1.1 kcal/molo (see page 12).

Reviewer #1 (Remarks to the Author):

The authors have addressed all my concerns in the revised manuscript.

Reviewer #2 (Remarks to the Author):

My comments have been addressed satisfactorily.